# Diverse Biological Processes Contribute to Transforming Growth Factor β-Mediated Cancer Drug Resistance

**DOI:** 10.3390/cells14191518

**Published:** 2025-09-28

**Authors:** James P. Heiserman, Rosemary J. Akhurst

**Affiliations:** 1Helen Diller Family Comprehensive Cancer Center, University of California San Francisco (UCSF), San Francisco, CA 94143, USA; 2Department of Anatomy, Institute of Human Genetics, University of California San Francisco (UCSF), San Francisco, CA 94143, USA; 3Institute of Regenerative Medicine, University of California San Francisco (UCSF), San Francisco, CA 94143, USA

**Keywords:** cancer, drug resistance, transforming growth factor beta

## Abstract

Therapy resistance is a major obstacle to cancer treatment, and transforming growth factor-beta (TGF-β) signaling has emerged as a major instigator across many cancer types and therapeutic regimens. Solid tumors overexpress TGF-β ligands, and canonical and non-canonical TGF-β signaling pathways drive molecular changes in most cell types within the tumor to hijack therapeutic responses. Cancer therapies further stimulate TGF-β release to potentiate this problem. Molecular mechanisms of TGF-β action supporting resistance include upregulation of drug efflux pumps, enhanced DNA Damage Repair, elaboration of stiffened extracellular matrix, and decreased neoantigen presentation. TGF-β also activates pro-survival pathways, such as epidermal growth factor receptor, B-cell lymphoma-2 expression, and AKT-mTOR signaling. TGF-β-induced epithelial-to-mesenchymal transformation leads to tumor heterogeneity and acquisition of stem-like states. In the tumor microenvironment, TGF-β induces extracellular matrix production, contractility, and secretion of immunosuppressive cytokines by cancer-associated fibroblasts that contribute to drug resistance. TGF-β also blunts cytotoxic T and NK cell activities and stimulates recruitment and differentiation of immunosuppressive cells, including T-regulatory cells, M2 macrophages, and myeloid-derived suppressor cells. The importance of TGF-β signaling in development of drug resistance cannot be understated and should be further explored mechanistically to identify novel molecular approaches and combinatorial drug dosing strategies to prevent drug-resistance.

## 1. Perspectives

Numerous molecular studies on the mechanisms of cancer drug resistance have identified components of the transforming growth factor-beta (TGF-β) signaling pathway as drivers of drug resistance to chemotherapy and immunotherapy in mouse models and in clinical data.

Multiple biological processes and molecular mechanisms contributing to TGF-β-driven cancer drug resistance have been reported, including promotion of epithelial-to-mesenchymal transformation (EMT) and stemness, activation of cancer-associated fibroblasts, and other potent immunosuppressive activities on and by diverse immune cell types.

Efforts to target TGF-β in the clinic have had limited success, in part due to activities on normal cells. Further study of interacting pathways and the critical cell types involved are warranted to develop nuanced therapies that target TGF-β driven drug resistance mechanisms, with the prospect for new drug designs, optimized combinatorial drug dosing and scheduling, and early intervention for prevention of drug resistance.

## 2. Introduction

The development of cancer drug resistance is an intractable problem for oncologists and the major cause of cancer deaths. Tumors may be intrinsically drug-resistant, or resistance may be acquired in response to therapeutic treatment. Resistance to therapies that target one specific kinase or kinase pathway may develop by somatic mutation of the drug target or recruitment of alternative signaling pathways that bypass that target [1,2,3,4]. However, apart from these highly target-specific molecular mechanisms, innate or acquired drug resistance may occur through more generalized mechanisms resulting from the heterogeneity and plasticity of tumor cells and their microenvironment. In the early 1990s, there was evidence that elevated transforming growth factor-β (TGF-β) contributes to tamoxifen resistance in breast cancer [5,6,7], and in 2012, using an RNAi screening platform, Huang et al. showed that loss of MED12 that normally suppresses expression of the type 2 TGF-β receptor (TGFBR2), induces chemotherapy resistance in a variety of cancer cell lines originating from distinct organ sites [8]. Such TGF-β signaling-mediated resistance was demonstrated in response to chemotherapy agents such as cisplatin and 5-fluorodeoxyuridine, as well as to targeted therapeutics including the tyrosine kinase inhibitors crizotinib, gefitinib, and erlotinib [8].

Acquisition of drug resistance develops not only in response to chemotherapy, but also to radiotherapy or immunotherapy. This can evolve gradually through two stages; The first stage involves alterations in tumor cell plasticity, such as partial epithelial-to-mesenchymal transformation (EMT), upregulation of stem cell markers and stress responses, and downregulation of antigen presentation machinery [9,10,11]. The second phase occurs in response to long-term cytotoxic treatment that debulks the tumor but leads to persistence of epigenetically altered drug-tolerant persister cells (DTPs) that may remain dormant in the body for years [12,13,14,15]. Elevated TGF-β signaling (Figure 1) is often a major contributor to the first phase in this process which is the focus of this short review. Here, we will emphasize the potential importance of targeting TGF-β signaling early during disease treatment to prevent the development of DTPs.

## 3. TGF-β-Stimulated Epithelial-to-Mesenchymal Transformation Drives Chemotherapy Resistance Through Multiple Mechanisms

TGF-β ligands come in three different varieties (TGF-β1, -β2, and -β3) that form a sub-family within a larger TGF-β “super family” which includes bone morphogenic proteins (BMPs), activins, inhibins, growth and differentiation factors (GDFs), and other ligands [22]. We focus here on signaling from the TGF-β sub-family, and we utilize “TGF-β” to signify any member of this subfamily (TGF-β1, -β2, and/or -β3), unless otherwise specified.

TGF-β is a major inducer of epithelial-to-mesenchymal transformation (EMT) [23,24,25], the cellular process by which epithelial cells gradually reduce or switch off expression of epithelial markers and acquire mesenchymal characteristics [26] (Figure 2). EMT of tumor cells is not a simple biphasic switch but requires gradual molecular changes that can eventually cause complete loss of expression of epithelial markers and transition to a myofibroblast-like tumor cell [26]. During this change in cellular phenotype, multiple molecular changes in gene and protein expression occur, and epigenetic changes stabilize this phenotype. As a result, cells acquire several functions that can potentially suppress drug sensitivity. Oshimori et al. (2015) provided evidence that TGF-β signaling promotes tumor heterogeneity and elevates glutathione metabolism that protects cells from oxidative stress, thereby promoting cisplatin resistance in a murine squamous cell carcinoma (SCC) model [27]. In another study that utilized a triple transgenic mouse model to track EMT in lung cancer cells in vivo, cells that had transitioned to a mesenchymal state were found to be resistant to cyclophosphamide, and in vitro experiments showed that TGF-β stimulation increased the number of cells that became mesenchymal, further increasing the links between TGF-β, EMT, and chemotherapy resistance [28].

TGF-β-induced EMT causes elevated expression of ATP-binding cassette (ABC) transporters that encode drug efflux pumps which lower intracellular drug concentrations. Expression of TGF-β is frequently elevated in the tumor and its microenvironment to drive activation of pSMAD2/3/4 signaling, and the nuclear SMAD2/3/4 complex binds and activates genes encoding a variety of ABC proteins. In triple-negative breast cancer, resistance to a Cyclin-Dependent Kinase type 7 (CDK7) inhibitor caused elevated phospho-SMAD3 binding to, and activation of, the *ABCG2* gene, and drug resistance was reversed by genetic knockdown of *ABCG2*, pharmacological blockade of TGF-β type 1 receptors, or downregulation of SMAD4 [29]. In ovarian cancers, TGF-β signaling has been shown to upregulate the homeobox transcription factor PITX2A/B, which binds and upregulates *ABCB1*, resulting in chemotherapy resistance [30]. In A549 lung cancer cells, da Costa et al. (2023) found that TGF-β-induced EMT enhanced the expression of the ABCB1 and ABCC1 drug efflux pumps, as well as increased the cellular detoxification activity of the ABCC1 pump [31]. In pancreatic ductal adenocarcinoma (PDAC), *ABCC1* expression is also elevated by TGF-β induction of the transcription factor ATF4 that binds the ABCC1 gene promoter [32] (Figure 2). Notably, ABCC1 exports glutathionylated substrates, causing resistance of various malignancies to anti-cancer drugs, including gemcitabine and cisplatin [32,33,34].

EMT can drive tumor cell resistance not only to chemotherapies but also to immunotherapies. TGF-β-induced EMT was shown to downregulate multiple components of the tumor cell neo-antigen presentation machinery, including *MHC1* itself, as well as downregulation of genes encoding β-2 microglobulin and transporter associated with antigen processing (TAP1 and TAP2), that process tumor antigens for presentation on the tumor cell surface [35]. More details on mechanisms recently came from studies in non-small cell lung cancer (NSCLC) undertaken by Tan et al., (2024) [36], showing that increased TGF-β signaling caused phosphorylation of UHRF1 (ubiquitin-like with plant homeodomain and ring finger domains 1). This resulted in mislocalization of UHRF1 within the cytoplasm, causing binding to MHC1 that is consequently ubiquitinated and degraded through the proteosome [36]. Inhibition of TGF-β may therefore potentiate immunotherapies, including immune checkpoint blockade (ICB), not only by activating immune cells directly, but by enhancing the visibility of the tumor cell to the adaptive immune system by elevating antigen presentation (Figure 2).

## 4. Induction of Drug-Resistant Cancer Stem Cells by TGF-β

EMT, or at least a partial EMT, has been postulated as a route for tumor cells to acquire the properties of cancer stem cells (CSCs) [10]. The cancer stem cell theory proposes that tumor cells arrange themselves into hierarchies within the tumor microenvironment (TME), at the top of which reside unique CSC clones that populate the tumor with their heterogeneous progeny. Critical to CSC theory is that CSCs are required to seed new tumors, implicating CSCs as a major driver of tumor progression and metastasis [10]. CSCs are also thought to contribute to chemotherapy resistance [37], although exactly how this happens is still under investigation.

In recent times, Katsuno et al. (2019) found that chronic TGF-β ligand stimulation resulted in a stabilized EMT in immortalized human mammary epithelial cells which included a stem cell-like state and resistance to the commonly employed chemotherapy agents, doxorubicin, cisplatin, and cyclophosphamide [38]. The authors further found that chronic TGF-β exposure led to this CSC-like phenotype through upregulation of AKT-mTOR signaling, and pharmacological mTOR inhibition attenuated TGF-β-stimulated chemoresistance [38]. Other studies identified additional mechanisms by which TGF-β signaling contributes to the CSC state in different organ-specific cancer sites. Loss of the tumor suppressor, *MYOCD*, enhances the stemness of NSCLC, which is partly due to loss of PRMT5/MEP500-mediated epigenetic silencing of the *TGFBR2* gene. MYOCD binds to the *TGFBR2* gene promoter and recruits PRMT5/MEP500 [39]. Similarly, upregulation of the type 1 TGF-β receptor is implicated in driving the CSC state and chemotherapy resistance in hepatocellular cancer. A CRISPR activation screen identified *TGFBRAP1*, encoding a TGFBR1 binding protein, as supporting hepatocellular CSC formation and drug resistance. TGFBRAP1 stabilizes TGFBR1 by competing with the E3 ubiquitin ligase SMURF, thus blocking TGFBR1 polyubiquitination and degradation. Moreover, a positive feedback loop stabilizes the CSC phenotype through upregulation of *TGFBRAP1* by TGF-β signaling [40]. Elevated TGF-β signaling through both SMAD and non-SMAD pathways (Figure 1) also increases expression of *PITX2* to drive stemness in ovarian cancer [30].

Taken together, TGF-β signaling may promote cancer resistance to chemotherapy by inducing tumor heterogeneity and cellular plasticity of cancer cells through EMT cellular programs, and EMT may increase the pool of CSCs while also providing pro-survival signaling within cancer cells, such as stimulation of the AKT-mTOR pathway, that increases tumor resistance to commonly used chemotherapy therapies in the clinic.

## 5. TGF-β-Induced Pro-Survival Signaling in Tumor Cells Is Elicited Through SMAD and Non-SMAD Pathways

Although there are certainly anti-proliferative and pro-apoptotic effects associated with TGF-β signaling in normal epithelial cells [41,42,43], it seems that at least some carcinomas dispose of this negative growth regulatory arm (due to oncogene activation or loss of tumor suppressors) and take advantage of the pro-survival signaling pathways triggered by the TGF-β receptor complex. *SMAD3* was recently identified in a CRISPR screen as a functional target for genes that promote MEKi resistance. An activated SMAD3 signature increased during development of BRAFi resistance and was high in DTPs and in tumor cells during relapse. SMAD3 appeared to drive resistance to MEKi in melanoma by directly upregulating the *EGFR* and *AXL* genes [44]. Nevertheless, the outcome of TGF-β action on driving apoptosis versus cell survival is context-dependent, and a recent study demonstrated that elevated TGF-β1 can synergize with MEK inhibition to stimulate apoptosis in melanoma [45].

As discussed above, AKT signaling plays a prominent role in terms of “non-canonical” TGF-β signaling pathways that directly promote cell survival downstream of TGF-β ligand/receptor engagement [38]. TGF-β stimulates AKT signaling through PI3K activation, and PI3K and AKT inhibitors have been extensively tested in pre-clinical models. These drugs, that are currently being assessed in clinical trials [46], may be applicable in reducing drug-evading effects elicited by excess TGF-β [47]. A recent report [48] suggests that in PDAC, TGF-β-induced AKT activation causes the accumulation of specific neutral lipid species that contribute to gemcitabine resistance. Another study showed that inhibition of TGF-β2, using the drug imperatorin, a naturally occurring furanocoumarin, led to synergistic tumor killing when used with gemcitabine in both in vitro and in vivo settings [49].

TGF-β signaling may support other cellular pro-survival factors to drive chemotherapy resistance, such as by upregulation and maintenance of elevated B-cell lymphoma 2 (BCL-2) expression in bone metastatic prostate cancer [50]. The authors of this study found that TGF-β promotes resistance to docetaxel through acetylation of the transcription factor KLF5, which in turn increases *BCL2* gene transcription. They additionally found that TGF-β prevented docetaxel-induced degradation of BCL-2 [50].

To conclude, TGF-β also promotes cancer cell survival and chemotherapy resistance through SMAD and non-SMAD signaling pathways, as well as through upregulation of proteins, such as EGFR, AXL and Hedgehog signaling from which there may be many downstream cellular effects that promote chemoresistance, such the acetylation of KLF5 that causes increased *BCL2* expression, or the accumulation of lipid species that promote cancer cell chemotherapy resistance.

## 6. TGF-β Signaling Blockade, an Achilles Heel for Genotoxic Therapies Through Suppression of DNA Damage Repair Pathways

Radiotherapy and platinum-based chemotherapy kill tumor cells by inducing DNA damage including single- and double-strand DNA (dsDNA) breaks that lead to mutations, cytogenetic rearrangements, and ultimately to tumor cell death [51]. These therapeutic modalities also induce expression and activation of TGF-β1 [52,53] that can result in innate drug resistance, as well as induction of fibrosis that reinforces drug resistance in the tumor by providing a source of activated CAFs (see below). Notably, in response to DNA damage elicited by radiotherapy or chemotherapy, several DNA Damage Repair (DDR) pathways are activated that, in one way or another, limit DNA damage to suppress tumor cell death, ultimately reducing the efficacy of these genotoxic therapies [54]. Importantly, TGF-β signaling has been shown to protect the genome from DNA damage by potentiating several DDR pathways that are activated by dsDNA breaks [55,56,57]. BRCA1/Rad51-mediated homologous recombination repair (HRR) is potentiated by TGF-β-mediated upregulation of BRCA2 [58]. DNA repair through the Non-Homologous End-Joining (NHEJ) pathway is also stimulated by TGF-β through elevated expression and nuclear translocation of DNA ligase IV (LIG4) [59,60]. Finally, in the absence of TGF-β signaling, a less efficient Alt-EJ repair pathway is preferentially utilized via activity of LIG1 (DNA ligase 1), PARP1, and POLQ [58], accentuating the effects of these therapies [56,59,61,62].

## 7. TGF-β and Tumor Immunity

Since 2013, when the major breakthrough of the year announced by the journal *Science*, was immune checkpoint blockade (ICB) for cancer therapy [63], many labs have focused their efforts on identifying the mechanisms of resistance to such immunotherapies. Apart from a low mutation load, which is unfavorable to immunotherapy responsiveness, transcriptomics analyses found that markers of EMT, ECM, and TGF-β signaling are associated with ICB resistance [64,65,66]. Strong empirical evidence in preclinical models then showed that blockade of TGF-β signaling may alleviate resistance to immunotherapy through activity on multiple immune cell types within the tumor and draining lymph nodes [47]. Indeed, TGF-β is a potent immunosuppressor of cytotoxic CD8+ T cells and NK cells and tends to polarize CD4+ T helpers and myeloid cells away from an anti-tumor phenotype towards an immunosuppressive phenotype [47,67,68]. Consistent with the idea that TGF-β signaling stimulates immune-suppressing cell types, Dodagatta-Marri et al. (2019, 2021) [35,69] not only confirmed in their SCC model that anti-PD1 therapy enhances cytotoxic CD8+ T cell infiltration and activation, but notably also found an increased fraction of CD4+ T cells that are immunosuppressive FOXP3+CD25+ T regulatory cells. Logically, an anti-TGF-β blocking antibody was able to synergistically enhance anti-PD1 therapy through attenuation of this drug-induced increase in immunosuppressive intratumoral Tregs [35], and consequent further elevation of cytotoxic CD8+ T cell activity [69].

More recently, another group examined transcriptomes from recurrent gynecologic cancer patients treated with either anti-PD-1 or anti-PD-L1 antibodies. They also found a TGF-β-driven genetic signature that predicted immunotherapy resistance as indicated by reduced overall and progression-free survival [70]. As in previous studies in other cancers [47], the immune cell types that correlate with high TGF-β signaling in these ovarian cancers also correlated with worse overall survival, including T regulatory cells (Tregs), M2 macrophages, and eosinophils. Interestingly, in that study, naïve CD4+ T cells along with follicular helper T cells were associated with high TGF-β signaling signatures, possibly representing a TGF-β-mediated suppression of T cell differentiation, but the mechanisms by which naïve CD4+ T cells and follicular helper T cells contribute to poorer overall survival in response to ICB remain to be deciphered [70]. The plethora of mechanisms whereby TGF-β signaling dampens immune responses to ICB has been reviewed extensively [47,71,72,73,74,75], and readers are advised to refer to these publications for details of this important TGF-β activity in mediating immunotherapy resistance.

## 8. Contribution of TGF-β to Tumor Microenvironment (TME)-Driven Therapy Resistance

In addition to inducing tumor cell plasticity, cancer cell heterogeneity, and promoting cancer stemness through EMT, TGF-β has profound effects on the tumor microenvironment (TME). Studies utilizing small molecular inhibitors that target TGFBR1 have shown reduced tumor angiogenesis, enhanced vessel stability, and lowered interstitial tumor pressure in breast cancer, cutaneous SCC, HCC, and glioblastoma in response to TGF-β signaling inhibition [76,77,78,79,80]. TGF-β is also a major modulator of fibroblasts and their transformation into cancer-associated fibroblasts (CAFs) that impacts both the tumor cell [47,81,82] as well as recruitment and cellular functions of diverse immune cell types within the tumor [47,66,83,84].

Several laboratories reported that TGF-β-driven transcriptomic signatures associate with the presence of CAFs as well as with immunosuppressive blood cells, such as M2 macrophages, Tregs, and myeloid-derived suppressor cells (MDSCs). This signature associates with decreased responses to ICB treatment, such as atezolizumab (anti-PDL-1) or pembrolizumab (anti-PD-1), in patients with head and neck (HN) SCC [85], melanoma [86], urothelial carcinoma [66], gynecological cancers [70], and breast cancer [87].

Across many solid tumor types, CAFs are known to contribute to tumor progression [47] and chemotherapy resistance [88], which can at least partially be attributed to their secretion of TGF-β ligands [89]. CAFs are a major contributor to the elaboration of ECM (extracellular matrix) proteins within the TME [90], a feature stimulated by TGF-β ligands. High levels of ECM production generate physical and chemical barriers to immune cell infiltration [66,84], as well as physical tension within the tumor that correlates with elevated TGF-β signaling and chemotherapy resistance, as observed in gastric cancer [91]. Dominguez et al. (2020) found that TGF-β stimulated a major subset of CAFs, identified by expression of Leucine-Rich-Repeats-Containing Protein 15, LRRC15, in PDAC and other cancers, and a high TGF-β CAF signature correlated with poor patient survival in an immunotherapy clinical trial across six cancer types [92]. In a later study, the same group found that LRRC15+ CAFs promote tumor progression and resistance to anti-PDL1 treatment via suppressing effector T cell function in various mouse tumor models [93]. In line with these studies on CAFs in PDAC immunotherapy resistance, a subsequent study from an independent group found that CAFs in esophageal cancer may also play a role in immunotherapy resistance by upregulating their expression of programmed death ligand-1 (PD-L1) [94], which when ligated to T cell-PD-1 is known to induce T cell anergy and apoptosis, the premise for ICB therapies.

In ovarian cancer, which metastasizes to the peritoneal cavity, tumor-derived TGF-β induces osteopontin expression by mesothelial cells lining the peritoneal cavity. Secreted osteopontin in turn induces activation of the CD44 receptor, PI3K/AKT signaling, and ABC drug pumps in tumor cells, contributing to chemotherapy resistance. Notably, targeting osteoponin in ovarian cancer models improved cisplatin efficiency of tumor killing [33]. In concordance with these studies showing that CAFs and macrophages promote chemo- and immunotherapy resistance, a more recent study using co-culture experiments showed that CAFs and TAMs (tumor-associated macrophages) promote neuroblastoma tumor cell survival through complex parallel mechanisms. Secretion of IL-6 and s-IL6Rα by “mesenchymal stromal cells”, also known as MSCs/CAFs, promotes monocyte survival, whilst TGF-β secretion from MSCs and tumor cells differentiates monocytes toward M2 macrophages [95]. The authors demonstrated that all three cell types secrete TGF-β1 ligand, which in turn was shown to inhibit cytotoxic NK cell function in vitro [95]. M2 macrophages contribute to chemotherapy resistance as a source of TGF-β ligands, which in turn promote a cancer stem cell-like phenotype that confers cisplatin resistance as observed in studies on esophageal squamous cell carcinoma [96].

In melanoma, Xu et al., (2025) [84] found that CD133+PDL-1+ tumor stem-like cells, which synthesize higher TGF-β quantities than their CD133-PDL-1- counterparts, develop a local niche characterized by high ECM elaboration. This, in turn, prevents recruitment and killing of tumor cells by adoptively transferred macrophages due to prevention of macrophage migration through the stiffened ECM. Moreover, locally high TGF-β prevents full differentiation of macrophages to a phagocytic phenotype for tumor cell clearance [84]. Finally, a recent study has shown that TGF-β trapped in Neutrophil “NETs” (Neutrophil extracellular traps) promotes resistance to two different chemotherapy regimens (cisplatin or combinatorial adriamycin plus cyclophosphamide) [97]. The authors found that in breast carcinoma, neutrophil “NETs” are formed within the tumor in response to chemotherapy, due to activation of the inflammasome by ATP released from dying cancer cells. This causes release of IL-1β that drives NET elaboration. Latent TGF-β released by tumor cells gets trapped in NETs through binding to integrin αvβ1 and is activated by NET-trapped MMPs. Activated TGF-β released from the NETs thus drives drug resistance in the tumor cells [97].

Finally, it must be mentioned that there is a recent interest in drug resistance driven by delivery of TGF-β ligands through extracellular vesicles (EVs) [98,99,100]. EVs come in a variety of sizes and can originate from various cell types within a tumor including cancer cells, stromal cells, and immune cells [100]. They can carry nucleic acid and lipid species, as well as proteins, including TGF-β ligands and other TGF-β signaling components [98,99,100,101]. TGF-β ligands were found to be in excreted EVs derived from prostate cancer cell lines. Moreover, levels of EV-derived TGF-β correlated with poor prognosis and bone metastasis in a cohort of 19 castration resistant prostate cancer patients [98]. In line with this study, another group found in a cohort of 33 patients with non–small cell lung cancer, that circulating EV TGF-β isolated from the patient’s blood correlated with lack of response to immune-checkpoint inhibitors (pembrolizumab and nivolumab) as well as worse progression-free and overall survival [99]. Both studies suggest utility of EV-borne TGF-β as a prognostic marker for reduced response to therapy. As a final note, there may be other components of TGF-β signaling that are shuttled via EVs. One group found that breast cancer-derived EVs that contained TβRII led to exhaustion of CD8+ cytotoxic T cells [101]. Intriguingly, the group further demonstrated that the CD8+ T cells were able to take up TβRII+ EVs, suggesting that the enhanced TβRII receptor signaling in these cells led to their exhaustion. Further study of EVs that contain TGF-β ligand and receptor signaling components, as it relates to cancer therapy resistance, is warranted.

To conclude, clearly TGF-β signaling has a profound effect on non-malignant cell types in the tumor microenvironment through the transformation of fibroblasts into CAFs and multiple immunosuppressive effects on myeloid cells. These immunosuppressive cell types promote immune evasion and survival of malignant tumor cells through further secretion of TGF-β ligands, as in the case of M2 macrophages, thus attenuating the efficacy of both chemotherapies and immune therapies.

## 9. Conclusions

Recent studies, supported by publications from decades past, have clearly demonstrated that aberrant upregulation and/or activation of TGF-β signaling in many cancers contribute to resistance to chemotherapy, radiotherapy, and immunotherapy through a plethora of mechanisms (Figure 3). TGF-β ligands, acting on cancer cells, drive phenotypic as well as molecular changes, inducing cancer cell heterogeneity and a drug-resistant phenotype that is associated with EMT, stemness, upregulation of drug efflux pumps, support of DNA Damage Repair mechanisms, pro-survival signaling, as well as decreased neoantigen presentation. Furthermore, TGF-β also acts on tumor-associated immune and non-immune cell types alike, providing protection of cancer cells from cytotoxic immune cells by elaboration of a stiff extracellular matrix and polarization of TME and immune cells towards immunosuppressive phenotypes (Figure 2). These studies have highlighted TGF-β signaling as a major driver of resistance to currently used cancer therapies and warrant the further exploration of the molecular and cellular consequences of TGF-β stimulation of cancer and stromal cells with the goal of discovering and designing new cancer treatment schemes.

## 10. Future Directions

Targeting TGF-β signaling in cancer continues to be of interest in drug development, but more nuanced approaches are required due to the widespread pleiotropic activities of TGF-β in many tissues and cell types. Targeting components of the TGF-β pathway that show more restricted expression, such as integrin β8 which is expressed at the highest levels in intratumoral CD4+CD25+ Tregs and tumor cells per se, may be one avenue towards success [69,102,103]. Another approach is the use of bispecific drugs that target TGF-β inhibition to specific cell types, such as a bispecific antibody composed of TGF-β-neutralizing TGFBR2 extracellular domain fused to ibalizumab, a non-immunosuppressive CD4 antibody [83], in order to specifically target TGF-β signaling in CD4+ T cells. Bintrafusp alfa has so far shown limited success in the clinic; this bispecific drug blocks both PDL-1 and TGF-βs and, importantly, causes drug homing to the tumor [74]. Another consideration is to target proteins that synergize with the TGF-β pathway to drive drug resistance, for example, targeting S100A4 or NFAT5 to prevent PDAC resistance to the Kras inhibitor, G12Di-MRTX1133 [104], or the use of the SRC inhibitor, dasatinib, to target the SRC-SLUG-TGF-β2 chemoresistance pathway in triple-negative breast cancer [105]. Although less amenable as drug targets, inhibition of TGFBRAP1 or PRMT5/MEP500 to alleviate drug resistance in NSCLC [39], or ATF4 inhibition to relieve gemcitabine resistance in PDAC [32], has been proposed. Importantly, identifying and optimizing the best combinatorial therapies for use with TGF-β signaling blockade and determining the ideal sequential dosing strategies for each drug combination remain to be fully investigated. Although difficult to achieve in a clinical trial setting, the use of combinatorial TGF-β blockade early during first-line therapy may be important to realize the full potential of blocking this pathway to prevent development of drug resistance.

## Figures and Tables

**Figure 1 cells-14-01518-f001:**
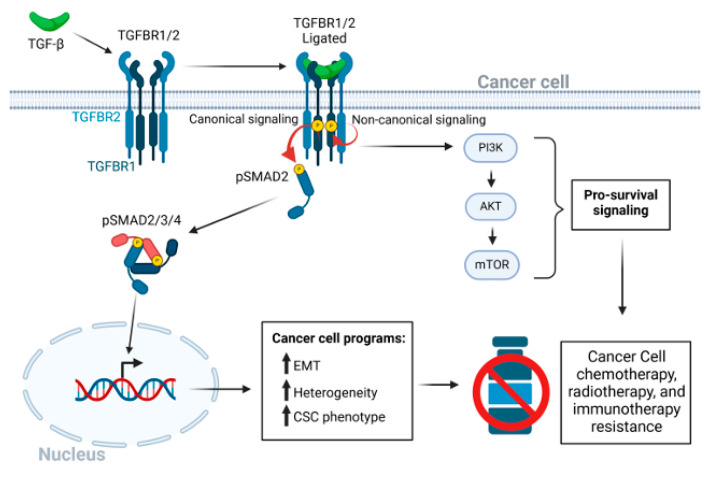
The TGF-β signaling pathway in cancer drug resistance. After ligation of dimeric TGF-β to the heterotetrametric TGF-β receptor complex of type 1 and type 2 receptors, a conformation change in TGFBR2 leads to kinase activation of TGFBR1. TGFBR1 kinase then activates the canonical receptor-associated SMAD2 and SMAD3 pathways through serine phosphorylation [16,17]. The activated hexameric SMAD2/3/4 complex enters the nucleus to regulate transcriptional programs that lead to EMT, CSC phenotypes, and tumor cell heterogeneity. Various non-canonical pathways may be triggered independent of the serine-threonine kinase activity of TGFBR1 [18,19], and both TGF-β receptors 1 and 2 have intrinsic tyrosine kinase activity, albeit less active than the canonical serine/threonine-specific activity [20,21]. Activation of the PI3K/AKT/mTOR signaling axis, independent of SMADs, promotes cancer cell survival. Red circular arrows indicated the serine threonine TGFBR-SMAD phosphorylation cascade. Thin black arrows indicate downstream consequences. Short black bold arrows indicate upregulation of processes.

**Figure 2 cells-14-01518-f002:**
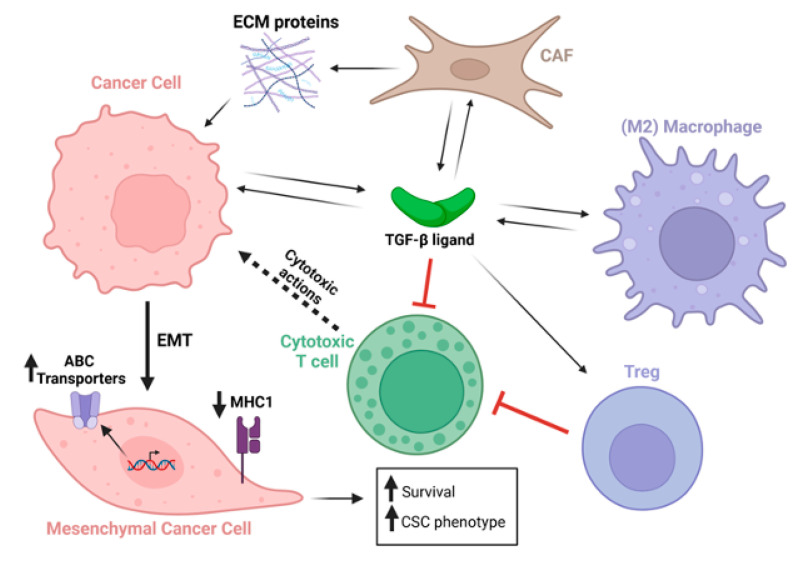
TGF-β secreted from multiple cell types acts on a plethora of cell types within the tumor and tumor microenvironment. TGF-β-stimulated cancer cells undergo EMT, which upregulates ABC transporters, downregulates MHC type 1 expression, and increases cancer cell survival and expression of cancer stem cell genes. Black arrows indicate secretion of TGF-β ligands, ECM proteins, or actions of these factors on cells. Dotted black arrow indicates cytotoxic activity emanating from cytotoxic T cells. Red connectors indicate inhibitory activity. Long bold black arrow indicates cancer cell epithelial-to-mesenchymal transformation (EMT). Short bold black arrows indicate upregulation or downregulation as indicated.

**Figure 3 cells-14-01518-f003:**
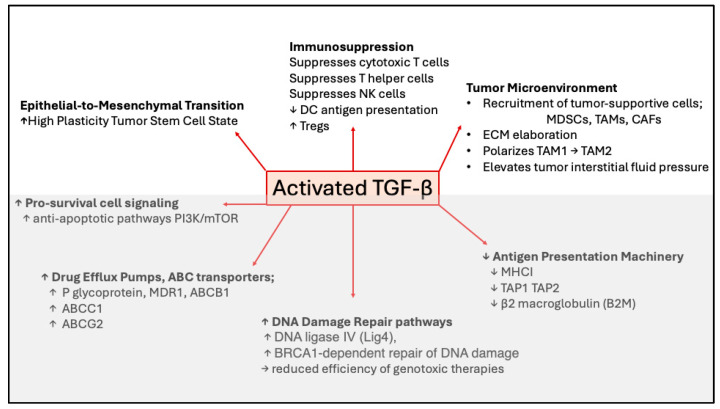
Mechanisms of TGF-β-mediated drug resistance. Cartoon showing mechanisms whereby TGF-β promotes resistance to chemotherapy, radiotherapy, or immunotherapy. Short upward pointing arrows indicate increases and short downward pointing arrows indicate decreases in cell type, process or protein to which the arrow refers. EMT, immunosuppression, and tumor microenvironment (shown in the upper figure) indicate TGF-β-induced changes in cell states; whereas in the lower figure (in gray) specific molecular mechanisms affected by TGF-β are indicated.

## Data Availability

Data sharing is not applicable.

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
