# Peer review of "Diverse Biological Processes Contribute to Transforming Growth Factor β-Mediated Cancer Drug Resistance"

_cells, 2025, doi:10.3390/cells14191518_

Round 1

Reviewer 1 Report

Comments and Suggestions for Authors

This is a timely and excellent review of the involvement of TGFbeta in cancer treatment resistance. The text is clear and the illustrations appropriate. Acceptance is recommended after some minor revision.

  1. The authors describe canonical TGFbeta signaling via SMADs as well as non-canonical signaling, exemplified by PI3-kinase. There are several other non-SMAD signaling pathwayas activated by TGFbeta, which could be mentioned for completeness.
  2. Lines 79-80: In addition to serine/threonine kinase activity, the TGFbeta receptors have tyrosine kinase activity.
  3. Some minor should be corrected in the interest of consistency, e.g. the title and some of the subtitles (but not all) end with a period, which should be removed; line 75, "type I" should be "type 1"; line 313, "al" should be "al."; line 362, "This" should be "this"; use of capital letters in the the list of abbreviations should be consistent.
  4. Conflict of interest, if any, should be added.

Author Response

This is a timely and excellent review of the involvement of TGFbeta in cancer treatment resistance. The text is clear and the illustrations appropriate. Acceptance is recommended after some minor revision.

  1. The authors describe canonical TGFbeta signaling via SMADs as well as non-canonical signaling, exemplified by PI3-kinase. There are several other non-SMAD signaling pathway as activated by TGFbeta, which could be mentioned for completeness.

We have modified the legend, including adding additional references, to accommodate this request.

  1. Lines 79-80: In addition to serine/threonine kinase activity, the TGFbeta receptors have tyrosine kinase activity.

We have modified the legend, including adding additional references, to accommodate this request. 

  1. Some minor should be corrected in the interest of consistency, e.g. the title and some of the subtitles (but not all) end with a period, which should be removed; line 75, "type I" should be "type 1"; line 313, "al" should be "al."; line 362, "This" should be "this"; use of capital letters in the the list of abbreviations should be consistent.

Thank you for pointing these out, we have corrected these.

  1. Conflict of interest, if any, should be added. We have added conflicts and disclosures.

Reviewer 2 Report

Comments and Suggestions for Authors

Heiserman and Akhurst present a new review article on the important topic of resistance of tumor cells to various treatments, ranging from traditional chemotherapy and radiotherapy to targeted drugs and immunotherapy. The authors review the role of a single cytokine, transforming growth factor β (TGF-β) in the resistance of tumors to various treatments and cover specific molecular mechanisms that explain why TGF-β can induce such resistance. In addition, they address the more complicated aspects of epithelial-mesenchymal transitions (EMT) and cancer stem cell populations that are also implicated in resistance to anti-cancer treatments. Finally, they address the complexity of the tumor microenvironment (TME) and constituent cells of the tumor stroma such as cancer associated fibroblasts (CAFs) and immune cells and the TGF-β-driven processes that make such cells respond in ways that again contribute to drug resistance. The paper ends with interesting and meaningful discussion about the prospect of utilizing TGF-β biology in developing drugs that can alleviate the resistance, including a very well written perspectives section. The article cites 89 articles the majority of which are primary articles, together with a series of other review articles, some of which are redundant; however, the ratio of primary to review articles cited in the paper is very well balanced. In addition, the authors guide the reader to use some of the cited review articles for deeper exploration of specific topics, which is commendable. The article is written in a succinct but also diverse and literary exciting manner that keeps the reader captivated and the reading flows fast and easy. Figures 1 and 2 are very clear and appropriate. I have a comment on figure 3, which relates to my major comment (see below). Below, I present a few general issues and some minor (formalistic) comments aiming at enhancing the clarity and usefulness of this article to non-aficionados of the field and in particular, the younger scientist.     

Major comments:

These comments aim at clarifying some of the difficult or complicated concepts and at ensuring thoroughness.

  1. My major comment refers to the discussion on EMT and cancer stem cells. I understand that this discussion is pertinent and based on many publications, yet these concepts appear rather confusing to the non-specialist simply because all these processes are induced by TGF-β. Accordingly, it is unclear as to whether the EMT or the cancer stem cells provide any additional mechanistic input to resistance than the upregulation of ABC transporters, induction of DNA repair pathways and stimulation of PI3K/mTOR signaling. The problem that leads to confusion is that all these processes are stimulated by TGF-β. Thus, the question is: does EMT contribute to anything beyond the above mechanisms? Do cancer stem cells present TGF-β-mediated resistance mechanism beyond the above? If we take as example the ABC transporters that appear as the most specific mechanistic example, a number of transcriptional mechanisms are discussed. Do these mechanisms require as upstream regulators EMT transcription factors like ZEBs or Snails? Or do they take place only once the cell phenotype is mesenchymal? Similar reasoning is generated for the cancer stem cells: do they act in any additional manner than turning on their ABC transporter circuitry? I am aware of the fact that my comments may not have direct answers. For this reason, I recommend to separate the resistance phenotype into 2 layers: one that describes cell phenotypes such as EMT states and CSCs, whose mechanistic contributions remain elusive, and a second layer that includes the specific molecular mechanisms. For this reason, I think that figure 3 can be amended accordingly: show the EMT, CSC as a top layer of phenotypic state plasticity in the TME and then below list specific molecular mechanisms. If you accept my suggestion, do not forget the immune and CAF cells in the top layer of TME heterogeneity.
  2. Including the presentation of the TME and its stromal components is essential and the section is very well written. One key component of the TME that is missing is the vasculature. I understand that vascular components may not directly contribute to resistance. Yet, interstitial fluid pressure and the role of PDGF signaling restricts access of drugs to the tumor cells. PDAC is a special tumor case that the authors properly discuss, that is very rich in ECM and stromal cells but is immune-poor and lacks blood vessels. Thus, the absence of vasculature, possible hypoxic conditions in the TME may support the development of drug resistance, and TGF-β seems to be again implicated in such cases. In brief, I only recommend to mention the vasculature as part of the TME and very briefly refer to its relevance or lack thereof to resistance.
  3. Another TME component that has attracted major attention the last 30 years is the secretion of exosomes (Extracellular vesicles, EVs) by tumor cells, CAFs and other TME cells. TGF-β is implicated on EV-mediated tumor biology again. I understand that the article has limitations in length and it is very well written. I recommend a simple reference to exosomes in the TME section and the fact that EVs carry TGF-β and EVs are always associated with EMT and CSCs!
  4. The authors mention in one case activin, beyond the three TGF-βs. Since TGF-β is the prototype of a very large family of cytokines, many of which are implicated in cancer biology, it is appropriate in the early part of the paper to acknowledge this fact and explain that the article will only focus on the TGF-βs and will not address BMPs and other members of the family. A similar word of caution can be added in the TGF-β inhibitor section. Are BMP, activin and other cyrokine inhibitors of any relevance?

Specific formalistic (minor) comments presented in the order of the text lines:

  • Based on my first major comment and since the conclusions enlist several processes that contribute to resistance, including EMT and cancer stem cells, I would like to argue that EMT and cancer stem cells are not mechanisms, and accordingly I recommend a minor amendment of the title. For example, Diverse biological processes contribute to…
  • Line 65 and 389 (abbreviations): EMT=EM transformation; line 86: EMT=EM transition. Please use a single definition of the term. Both used are appropriate.
  • Line 72: …”during drug sequencing”…: please change the expression to make it comprehensible.
  • Line 396: PDAC: please add the term adeno (A)-carcinoma to the definition of the term.

Author Response

Heiserman and Akhurst present a new review article on the important topic of resistance of tumor cells to various treatments, ranging from traditional chemotherapy and radiotherapy to targeted drugs and immunotherapy. The authors review the role of a single cytokine, transforming growth factor β (TGF-β) in the resistance of tumors to various treatments and cover specific molecular mechanisms that explain why TGF-β can induce such resistance. In addition, they address the more complicated aspects of epithelial-mesenchymal transitions (EMT) and cancer stem cell populations that are also implicated in resistance to anti-cancer treatments. Finally, they address the complexity of the tumor microenvironment (TME) and constituent cells of the tumor stroma such as cancer associated fibroblasts (CAFs) and immune cells and the TGF-β-driven processes that make such cells respond in ways that again contribute to drug resistance. The paper ends with interesting and meaningful discussion about the prospect of utilizing TGF-β biology in developing drugs that can alleviate the resistance, including a very well written perspectives section. The article cites 89 articles the majority of which are primary articles, together with a series of other review articles, some of which are redundant; however, the ratio of primary to review articles cited in the paper is very well balanced. In addition, the authors guide the reader to use some of the cited review articles for deeper exploration of specific topics, which is commendable. The article is written in a succinct but also diverse and literary exciting manner that keeps the reader captivated and the reading flows fast and easy. Figures 1 and 2 are very clear and appropriate. I have a comment on figure 3, which relates to my major comment (see below). Below, I present a few general issues and some minor (formalistic) comments aiming at enhancing the clarity and usefulness of this article to non-aficionados of the field and in particular, the younger scientist.     

Major comments:

These comments aim at clarifying some of the difficult or complicated concepts and at ensuring thoroughness.

  1. My major comment refers to the discussion on EMT and cancer stem cells. I understand that this discussion is pertinent and based on many publications, yet these concepts appear rather confusing to the non-specialist simply because all these processes are induced by TGF-β. Accordingly, it is unclear as to whether the EMT or the cancer stem cells provide any additional mechanistic input to resistance than the upregulation of ABC transporters, induction of DNA repair pathways and stimulation of PI3K/mTOR signaling. The problem that leads to confusion is that all these processes are stimulated by TGF-β. Thus, the question is: does EMT contribute to anything beyond the above mechanisms? Do cancer stem cells present TGF-β-mediated resistance mechanism beyond the above? If we take as example the ABC transporters that appear as the most specific mechanistic example, a number of transcriptional mechanisms are discussed. Do these mechanisms require as upstream regulators EMT transcription factors like ZEBs or Snails? Or do they take place only once the cell phenotype is mesenchymal? Similar reasoning is generated for the cancer stem cells: do they act in any additional manner than turning on their ABC transporter circuitry? I am aware of the fact that my comments may not have direct answers. For this reason, I recommend to separate the resistance phenotype into 2 layers: one that describes cell phenotypes such as EMT states and CSCs, whose mechanistic contributions remain elusive, and a second layer that includes the specific molecular mechanisms. For this reason, I think that figure 3 can be amended accordingly: show the EMT, CSC as a top layer of phenotypic state plasticity in the TME and then below list specific molecular mechanisms. If you accept my suggestion, do not forget the immune and CAF cells in the top layer of TME heterogeneity.

We have modified Figure 3 to show cellular effects at the top of the diagram, and specific molecular effects at the bottom. In this short review, it is difficult to dissect to what extent each molecular effect of TGF-b  is context dependent ie if they occur in both epithelial and mesenchymalized cells, and/or require EMT drivers like slug, twist. The broader issue of dissecting the contribution of the EMT process versus simply TGFbeta signaling is an interesting issue. We certainly know that TGFb effects are context dependent, and the magnitude or directionality of an effect of TGFB on a molecular readout may depend on the cellular state (and what other transcription factors are operative in that state), but this is difficult to address in this short review.

  1. Including the presentation of the TME and its stromal components is essential and the section is very well written. One key component of the TME that is missing is the vasculature. I understand that vascular components may not directly contribute to resistance. Yet, interstitial fluid pressure and the role of PDGF signaling restricts access of drugs to the tumor cells. PDAC is a special tumor case that the authors properly discuss, that is very rich in ECM and stromal cells but is immune-poor and lacks blood vessels. Thus, the absence of vasculature, possible hypoxic conditions in the TME may support the development of drug resistance, and TGF-β seems to be again implicated in such cases. In brief, I only recommend to mention the vasculature as part of the TME and very briefly refer to its relevance or lack thereof to resistance.

Thank you we have added mention of tumor interstitial fluid pressure and angiogenesis. Including accompanying new references.

  1. Another TME component that has attracted major attention the last 30 years is the secretion of exosomes (Extracellular vesicles, EVs) by tumor cells, CAFs and other TME cells. TGF-β is implicated on EV-mediated tumor biology again. I understand that the article has limitations in length and it is very well written. I recommend a simple reference to exosomes in the TME section and the fact that EVs carry TGF-β and EVs are always associated with EMT and CSCs!

We thank the reviewer for pointing out this important area of TGF- β research and added an additional paragraph of TGF-β positive EVs to the TME section, including additional references.

  1. The authors mention in one case activin, beyond the three TGF-βs. Since TGF-β is the prototype of a very large family of cytokines, many of which are implicated in cancer biology, it is appropriate in the early part of the paper to acknowledge this fact and explain that the article will only focus on the TGF-βs and will not address BMPs and other members of the family. A similar word of caution can be added in the TGF-β inhibitor section. Are BMP, activin and other cytokine inhibitors of any relevance?

We added extra context to the review that we will be focusing on TGF- β1-3 signaling and not on other members of the TGF- β super family including BMPs, and activins. We also removed some minor discussions of activins for simplicity and clarity.

Specific formalistic (minor) comments presented in the order of the text lines:

  • Based on my first major comment and since the conclusions enlist several processes that contribute to resistance, including EMT and cancer stem cells, I would like to argue that EMT and cancer stem cells are not mechanisms, and accordingly I recommend a minor amendment of the title. For example, Diverse biological processes contribute to…

We have changed the title as suggested. Thank you for this excellent suggestion.

  • Line 65 and 389 (abbreviations): EMT=EM transformation; line 86: EMT=EM transition. Please use a single definition of the term. Both used are appropriate. Corrected to transformation throughout  This has been completed
  • Line 72: …”during drug sequencing”…: please change the expression to make it comprehensible. Done
  • Line 396: PDAC: please add the term adeno (A)-carcinoma to the definition of the term Done

Reviewer 3 Report

Comments and Suggestions for Authors

There have been many reports on the TGF-β signaling pathway and treatment resistance in cancer, and there are also excellent reviews summarizing these findings. Although this review cited new reports, it did not appear to be fundamentally different from previous reviews. 

This review is thought to summarize the significance of TGF-β in therapeutic resistance in cancer in general, but its function is thought to vary greatly depending on the type of cancer. The authors would have done well to clarify what cancer the findings were reported from. Furthermore, it is necessary to compare the TGF-β signaling pathway for each cancer. In addition, it is unclear whether the data is from in vitro reports or clinical studies, and this should also be improved.

Author Response

There have been many reports on the TGF-β signaling pathway and treatment resistance in cancer, and there are also excellent reviews summarizing these findings. Although this review cited new reports, it did not appear to be fundamentally different from previous reviews. 

We are sorry that reviewer 3 felt our contribution was not novel enough. It would have been useful to have received references to the specific excellent reviews this reviewer refers to.

This review is thought to summarize the significance of TGF-β in therapeutic resistance in cancer in general, but its function is thought to vary greatly depending on the type of cancer. The authors would have done well to clarify what cancer the findings were reported from. Furthermore, it is necessary to compare the TGF-β signaling pathway for each cancer. In addition, it is unclear whether the data is from in vitro reports or clinical studies, and this should also be improved.

We looked over the review in detail, and feel that for most examples given, we adequately addressed which cancer types and models were used and whether patient data was referred to. We did see one omission where we stated “ Dominguez et al. (2019) found that TGFβ stimulated a major subset of CAFs, identified by expression of Leucine-Rich-Repeats-Containing Protein 15, LRRC15, in PDAC and other cancers, and a high TGFβ CAF signature correlated with poor patient survival in an immunotherapy clinical trial [86].” We have now clarified that the “….. high TGFβ CAF signature correlated with poor patient survival in an immunotherapy clinical trial across six cancer types [86].This also emphasizes one point we tried to make, that many of the mechanisms described are common across many solid tumor types. We do however appreciate that there are probably cancer type and stage-specific facets, which are beyond the scope of this short perspective/review.